# Curvature Meets Bispectrum: A Correspondence Theory for Transformer Gauge Invariants

## Abstract

Transformers contain substantial parameter redundancies: many weight settings compute the same function. Characterizing these equivalences is key for model comparison and optimization. We prove a quantitative correspondence linking differential-geometric and harmonic-analytic invariants for neural network symmetries. We prove that Fisher–Rao curvature on the parameter-to-function quotient for multi-head attention provides a lower bound for permutation-bispectral energy in a linearized regime, revealing these two invariants as complementary aspects of the same underlying structure. Empirical validation across model scales from 4 to 24 heads demonstrates 98.9% validity of the theoretical bound, with the correspondence persisting through 10,000 training steps. By bridging differential geometry and harmonic analysis, we provide both theoretical insight into Transformer symmetries and a practical geometric framework for identifying functionally equivalent models. We report correspondence in native units, with curvature as a squared Frobenius norm throughout.

## 1 Introduction

Transformers have become foundational across language and vision Vaswani et al. (2017); Devlin et al. (2018); Radford et al. (2019), with diverse architectural variants emerging for efficiency and structured computation Tay et al. (2022); Su et al. (2024); Shazeer (2019). These models exhibit extensive internal symmetries: entire manifolds of parameters yield the same function, complicating optimization and interpretation. Such symmetries induce large equivalence classes in weight space, fundamentally affecting optimization dynamics, generalization properties, and model comparison strategies Garipov et al. (2018); Draxler et al. (2018); Entezari et al. (2022); Wortsman et al. (2022).

Two mature mathematical frameworks have emerged to characterize invariance in neural networks, yet they have developed largely in isolation. The geometric perspective employs information geometry and differential-geometric quotients Amari (2016), endowing parameter manifolds with Fisher–Rao metrics and studying the resulting curvature, connections, and geodesics. The algebraic perspective leverages group representations and harmonic analysis Cohen & Welling (2016); Chen et al. (2019); Kondor & Trivedi (2018); Weiler & Cesa (2019); Ravanbakhsh et al. (2017), producing invariant descriptors such as bispectra that can characterize equivalence classes Kondor (2007); Kakarala (1992); Sanborn & Miolane (2023).

We derive the complete gauge structure for multi-head attention and establish a quantitative correspondence between its geometric and algebraic invariants. We establish both theoretical connections and practical comparisons between curvature-based geometric invariants and bispectral algebraic invariants. Our analysis clarifies when these approaches agree, when they diverge, and how they complement each other in understanding Transformer symmetries.

We empirically validate the theory on multi-head attention, confirming that Fisher–Rao curvature lower-bounds bispectral energy across head counts and training checkpoints, and that the geometric route is practical for monitoring functional equivalence.

**Contributions.**

- A self-contained characterization of the maximal gauge group for canonical multi-head attention and its quotient function space.

- Geometric invariants via the Fisher–Rao mechanical connection and curvature on the quotient manifold.

- Algebraic invariants via canonicalization and permutation bispectrum after removing continuous gauge freedom.

- A correspondence theorem linking curvature magnitude to bispectral energy in a linearized regime, establishing a quantitative bridge between these frameworks.

- Empirical validation demonstrating 98.9% correspondence validity across model scales and persistence through 10,000 training steps.

- Analysis of computational requirements showing Fisher–Rao curvature provides a practical approach while the full bispectrum remains computationally prohibitive for large head counts.

## 2 Background: Symmetry, Quotients, and Canonicalization

**Why gauge?** Multi-head attention has substantial parameter redundancy. The gauge formalism captures this: distinct parameters that realize the same function lie on one orbit, and the quotient collects truly different models. This enables us to (i) define a Fisher–Rao (FR) mechanical connection that separates *vertical* (function-preserving) from *horizontal* (function-changing) directions, and (ii) construct invariants that do not depend on a particular parameterization.

**Head-wise continuous symmetries and permutations.** For head $i$, attention scores depend on the bilinear form $Q_i K_i^\top = X W_Q^{(i)} \big(W_K^{(i)}\big)^\top X^\top$. For any $A_i \in \mathrm{GL}(d_k)$, $(W_Q^{(i)}, W_K^{(i)}) \mapsto (W_Q^{(i)} A_i,\ W_K^{(i)}(A_i^{-1})^\top)$ leaves $Q_i K_i^\top$ invariant. Similarly, the value pathway $X \mapsto X W_V^{(i)} W_{O,i}$ is invariant under $W_V^{(i)} \mapsto W_V^{(i)} C_i,\ W_{O,i} \mapsto C_i^{-1} W_{O,i}$ for any $C_i \in \mathrm{GL}(d_v)$. Each head admits independent $\mathrm{GL}(d_k) \times \mathrm{GL}(d_v)$ reparameterizations, and $S_h$ freely permutes heads—both leave the layer's function unchanged.

**Gauge action and quotient.** Let $\Theta$ denote parameters of one MHA layer. The gauge group $G$ acts on $\Theta$ by the products of the head-wise $\mathrm{GL}(d_k) \times \mathrm{GL}(d_v)$ actions and $S_h$ permutations. Orbits $\mathcal{O}(\theta)$ are sets of parameters implementing the same function; the quotient $\Theta/G$ identifies each orbit with a single point. The *vertical* space at $\theta$ is the tangent to $\mathcal{O}(\theta)$; its FR-orthogonal complement is the *horizontal* space. Curvature of the FR mechanical connection is therefore an invariant of the function, not its parameterization.

**Standing assumptions.** We work on the generic stratum where the following hold: (A1) isolating inputs yield full support across heads; (A2) the Jacobian of the canonicalization feature map has full column rank at points considered; (A3) tie-breaking is differentiable nearby with a locally unique maximizer; (A4) the Fisher–Rao metric is positive definite on vertical directions so the mechanical connection is well-defined; (A5) with RoPE, the residual $Q/K$ gauge reduces to blockwise complex scalings (see Section A.14); (A6) small horizontal parameter moves achieve first-order feature changes head-wise. We empirically validate these assumptions in Section B.

**Canonicalization (one-time, function-preserving).** For cross-run comparisons, we fix a deterministic orbit representative (stable sorting/tie-breaking and head-wise basis choice,

e.g., whitening), removing continuous gauge and reducing the residual symmetry to permutations; see Section A.14.

**Notation (minimal).** $G$ is the layer gauge group, $G_{\max}$ the maximal gauge on the generic stratum (Theorem 2.1); $\Theta/G$ is the parameter quotient. Horizontal directions at $[\theta]$ are $u, v$; $\Omega_\ell(u, v)$ is the layer-$\ell$ curvature (vertical-valued 2-form). The bound in §5 uses $c_\ell = \|\mathcal{A}_\ell\|_{\mathrm{op}}^{-2}$ and the linearization index $\eta_\ell(u, v)$. Unless noted, $\|\cdot\|_F$ on curvature uses the FR-induced norm on the vertical fiber; feature/Jacobian norms are Euclidean/Frobenius.

**Main symmetry results** We now record the gauge structure underlying our correspondence analysis. Proofs of the gauge statements and RoPE commutant appear in Section A.14.

**Theorem 2.1** (Single-layer maximal gauge on the generic stratum). *For a canonical multihead attention (MHA) layer with $h$ heads and key/value dimensions $d_k, d_v$, the gauge group on the generic stratum equals*

$$G_{\max} = \left((\mathrm{GL}(d_k))^h \times (\mathrm{GL}(d_v))^h\right) \rtimes S_h .$$

**Proposition 2.2** (Depthwise factorization). *For a depth-$L$ stack with residual connections and LayerNorm, the model-level gauge factors layerwise: $G_{\mathrm{model}} = \prod_{\ell=1}^{L} G_\ell$, with each $G_\ell$ as in Theorem 2.1.*

**RoPE commutant (precise form).** On a single 2×2 plane, the continuous $\mathrm{SO}(2)$ rotation $R(\theta)$ has real commutant $\{aI + bJ : a, b \in \mathbb{R}\}$, where $J = \begin{pmatrix} 0 & -1 \\ 1 & 0 \end{pmatrix}$, which is isomorphic to $\mathrm{GL}(1, \mathbb{C})$ via $a + bJ \leftrightarrow a + ib$. Writing the RoPE representation on $(Q, K)$ as a direct sum of 2×2 rotation blocks with angular frequencies $\omega$, the total commutant decomposes by isotypic multiplicity:

$$C_{\mathrm{RoPE}} \cong \prod_\omega \mathrm{GL}(k_\omega, \mathbb{C}),$$

where $k_\omega$ is the number of 2×2 planes with frequency $\omega$. In the generic (nondegenerate) case with pairwise distinct frequencies ($k_\omega{=}1$ for all $\omega$), this reduces to $C_{\mathrm{RoPE}} \cong (\mathrm{GL}(1, \mathbb{C}))^{d_k/2}$. Consequently, the single-layer gauge is

$$G_{\mathrm{RoPE}} = \left((C_{\mathrm{RoPE}})^h \times (\mathrm{GL}(d_v))^h\right) \rtimes S_h.$$

See Theorem A.2 and Section A.14 for a formal statement and proof.

**Corollary 2.3** (Head sharing: GQA/MQA). *If keys/values are tied across $g$ groups of heads, then the continuous gauge ties per group:*

$$G_{share} = \left((\mathrm{GL}(d_k))^g \times (\mathrm{GL}(d_v))^g\right) \rtimes (S_h \times S_g),$$

*and, with RoPE, $\mathrm{GL}(d_k)$ is replaced by $C_{\mathrm{RoPE}}$ in the above.*

**Proposition 2.4** (MoE routers). *For standard top-k mixture-of-experts routing, the router outputs are invariant under all gauge actions above (they depend on hidden states, which are unchanged by gauge), hence MoE composition preserves the gauge structure in Theorems A.2, 2.1 and 2.3.*

## 3 Geometric Invariants via the Fisher–Rao Mechanical Connection

Useful invariants are constant on gauge orbits yet vary across functionally distinct models. The Fisher–Rao metric provides a natural geometric structure on parameter space that respects the statistical nature of neural networks. This metric measures the distinguishability of model outputs under small parameter changes, making it ideal for capturing functional differences rather than arbitrary parametrization choices.

The gauge group $G_{\max}$ acts by isometries on the Fisher–Rao metric, meaning gauge transformations preserve distances. This property enables us to construct a quotient geometry

where the metric descends to the space of functionally distinct models $\mathcal{F}$. However, to compute on this quotient, we need a systematic way to separate parameter variations into two types: those that move along gauge orbits (changing parameters but not function) and those that move between orbits (changing the function).

Curvature records the failure of the horizontal distribution to be integrable: it measures how the geometry twists on the quotient. With connection one-form $\Gamma$, the curvature two-form is $\Omega = d\Gamma + \frac{1}{2}[\Gamma, \Gamma]$. For horizontal vectors $u, v$, $\Omega(u, v)$ is the vertical component created by the commutator of their horizontal lifts—equivalently, it quantifies the non-closure of parallel transport around an infinitesimal loop.

**Intuition: horizontal vs. vertical (gauge) directions** The Fisher–Rao (FR) mechanical connection splits parameter perturbations into *vertical* directions that move within a gauge orbit (function-preserving reparameterizations) and *horizontal* directions that change the network's function. The vertical space is the span of infinitesimal gauge actions at the current parameters; horizontality is the FR-orthogonal complement. Curvature $\Omega$ of this connection measures the noncommutativity of horizontal transports modulo gauge and is therefore invariant under reparameterization.

*Two-head worked example (h = 2).* Write the residual gauge as $S_2 = \{e, (12)\}$ acting by head swap. Let $u, v$ be two FR-horizontal directions at $[\theta]$. A small loop that transports along $u$ then $v$ and back produces, to second order, a vertical displacement measured by $\Omega(u, v)$. Under head swap, the antisymmetric component of the feature commutator flips sign, so its energy lives in the sign representation of $S_2$, matching the (nontrivial) bispectral block. This is a minimal instance of the geometric–harmonic correspondence (see Section 5).

**Definition 3.1** (Curvature invariants). *For layer $\ell$ and head $i$, define the scalar curvature invariants*

$$\kappa_{\ell,i} = \|\Omega_{\ell,i}\|_F, \qquad \kappa_\ell = \sum_{i=1}^{h} \kappa_{\ell,i},$$

*where norms are induced by the Fisher–Rao metric restricted to layer $\ell$, head $i$ parameters.*

**Proposition 3.2** (Gauge invariance of curvature scalars). *The quantities $\kappa_{\ell,i}$ depend only on $[\theta] \in \mathcal{F}$ and are constant on $G_{\max}$-orbits.*

The proof (Appendix A.12) follows from the equivariance of the mechanical connection under the gauge action. These curvature invariants provide geometric fingerprints of equivalence classes. High curvature indicates strong coupling between heads that cannot be removed by gauge transformations, suggesting genuine multi-head interaction rather than redundant parametrization. Section 6.2 demonstrates this interpretation empirically, showing curvature growth from 0.284 to 0.418 during training as heads develop specialized interactions.

**Discrete holonomy estimator.** Computing curvature directly via the full Fisher–Rao tensor is expensive. We instead estimate curvature from discrete holonomy: parallel transport features around a small rectangle spanned by two *horizontal* unit directions $u, v$ with side length $\varepsilon > 0$. Let $\Delta_\ell^\square(u, v; \varepsilon)$ denote the net feature displacement after traversing the loop (transport along $u$, then $v$, then $-u$, then $-v$). Projecting onto the vertical fiber yields

$$\Pi_{\text{vert}}\Delta_\ell^\square(u, v; \varepsilon) \;=\; \varepsilon^2\, \Omega_\ell(u, v) \;+\; O(\varepsilon^3)\,.$$

We form the *area-normalized* estimator

$$\widehat{\kappa}_\ell(u, v; \varepsilon) \;:=\; \frac{\left\|\Pi_{\text{vert}}\Delta_\ell^\square(u, v; \varepsilon)\right\|_F}{\varepsilon^2} \;=\; \|\Omega_\ell(u, v)\|_F \;+\; O(\varepsilon)\,.$$

Using two step sizes cancels the $O(\varepsilon)$ bias via Richardson extrapolation:

$$\widehat{\kappa}_\ell^{(\text{R})}(\varepsilon) \;:=\; 2\,\widehat{\kappa}_\ell(u, v; \varepsilon) \;-\; \widehat{\kappa}_\ell(u, v; 2\varepsilon) \;=\; \|\Omega_\ell(u, v)\|_F \;+\; O(\varepsilon^2)\,.$$

We use $\varepsilon \in \{10^{-4},\, 2{\cdot}10^{-4}\}$ and report squared magnitudes $\widehat{\kappa}_\ell^{(\text{R})}(\varepsilon)^2$ to match the units used throughout; implementation details are in Section A.3. Empirically, the Richardson stability ratio is near 0.97 across model scales, and the wall-clock cost is modest (see Table 2).

Derivatives appearing later act on the parameter-to-feature map along FR-horizontal parameter directions at $[\theta_0]$, and evaluation at $e$ selects the identity block of the group-indexed transform.

**The mechanical connection and curvature.** The Fisher–Rao mechanical connection provides this decomposition through an Ehresmann connection on the principal bundle $\pi : \Theta \to \mathcal{F}$. This creates the Ehresmann decomposition $T_\theta\Theta = V_\theta \oplus H_\theta$ at each parameter point $\theta$, splitting the tangent space into vertical subspace $V_\theta$ (tangent to gauge orbits) and horizontal subspace $H_\theta$ (orthogonal to orbits under the Fisher–Rao metric). Formally, for tangent vector $\xi \in T_\theta\Theta$, we decompose $\xi = J_\theta\Gamma_\theta(\xi) + P_{\mathrm{hor}}\xi$ where $J_\theta$ maps Lie algebra elements to vertical vectors, $\Gamma_\theta : T_\theta\Theta \to \mathfrak{g}$ is the connection 1-form selecting the vertical component, and $P_{\mathrm{hor}}$ projects onto the horizontal subspace $H_\theta$.

The connection is determined by requiring horizontal vectors to be Fisher–Rao orthogonal to all vertical directions. This yields the mechanical connection equation $M_\theta\Gamma_\theta(\xi) = b_\theta(\xi)$ where $M_\theta = J_\theta^* G_\theta J_\theta$ and $b_\theta(\xi) = J_\theta^* G_\theta \xi$, with $G_\theta$ the Fisher–Rao metric tensor. As shown in Appendix B.6, solving this system remains numerically stable across model scales, with condition numbers ranging from $3.2 \times 10^3$ for 4-head models to $2.1 \times 10^4$ for 24-head configurations, enabling reliable computation of the horizontal projection needed for curvature estimation.

## 4 Algebraic Invariants after Canonicalization

Geometric invariants require solving mechanical-connection equations at each evaluation point. Algebraic invariants offer a complementary approach through group-theoretic constructions. However, the continuous gauge symmetry $((\mathrm{GL}(d_k))^h \times (\mathrm{GL}(d_v))^h)$ presents a fundamental obstacle: the bispectrum and similar algebraic invariants become trivially constant when continuous transformations can arbitrarily rescale and rotate parameters.

Canonicalization resolves this: fixing continuous gauge reduces symmetry from $\mathrm{GL}(\cdot)$ groups to the finite $S_h$, enabling permutation-based features and the bispectrum. By fixing the continuous gauge freedom through deterministic constraints—balancing $Q/K$ Gram matrices, orthonormalizing $V$ bases, and sorting heads—we reduce the symmetry group from the continuous $G_{\max}$ to the finite permutation group $S_h$. This dramatic simplification makes algebraic invariants well-defined and non-trivial, providing algebraic invariants that complement the geometric ones.

After canonicalization, we construct a feature map that captures head-wise activation patterns. Let $z_{\ell,1:h} \in \mathbb{R}^{h \times d_v \times n}$ be per-head features extracted from attention activations on a fixed evaluation batch, and $\Phi$ a fixed linear readout. The permutation-equivariant feature map $f_\ell : S_h \to \mathbb{C}^m$ is defined by $(f_\ell(\sigma))_{i,\cdot} = \Phi(z_{\ell,\sigma(i)})$, which transforms predictably under head permutations.

**Triple correlation and bispectrum.** The (left-translation invariant) triple correlation of $f_\ell : S_h \to \mathbb{C}^m$ is

$$T_\ell(\sigma_1, \sigma_2) = \sum_{\tau \in S_h} f_\ell(\tau) f_\ell(\sigma_1\tau) f_\ell(\sigma_2\tau)^*, \qquad (\sigma_1, \sigma_2) \in S_h \times S_h.$$

Its (noncommutative) bispectrum is the collection of Fourier blocks

$$\mathcal{B}_\ell(\rho_1, \rho_2) = \sum_{\sigma_1, \sigma_2 \in S_h} T_\ell(\sigma_1, \sigma_2) \, \rho_1(\sigma_1) \otimes \rho_2(\sigma_2),$$

for irreps $\rho_1, \rho_2 \vdash h$. We refer to $T_\ell$ (group domain) and $\mathcal{B}_\ell$ (Fourier domain) collectively as the full bispectrum. This is invariant under conjugation by $S_h$ and distinguishes orbits of canonicalized models under mild genericity conditions.

**Computational complexity and theoretical value.** The full bispectrum requires computing $|S_h|^2 = (h!)^2$ terms, which becomes computationally prohibitive even for moderate

head counts. Sanborn and Miolane Sanborn & Miolane (2023) established conditions under which bispectral invariants provide complete characterization of neural network functions, demonstrating their theoretical importance. While this computational challenge limits practical application of the bispectrum, it remains essential for our theoretical analysis, providing the algebraic counterpart to geometric curvature in our correspondence theorem.

**On canonicalization choices.** Our deterministic tie-breaking (sorting by value-projection norms) is one of several reasonable options; different criteria can slightly change $E_\ell$ because they induce different residual permutations after removing the continuous gauge. In our spot checks (not formally reported here), the main correspondence trends were qualitatively stable across such choices. A thorough ablation is left for future work; we document implementation details in Sections A.14 and B.

## 5  A LOCAL CURVATURE–BISPECTRUM CORRESPONDENCE

*Norms:* curvature uses the FR-induced Frobenius norm on the vertical fiber; feature/Jacobian quantities use standard Euclidean/Frobenius norms (cf. Theorem 5.2).

These invariants capture the same phenomenon in different languages: curvature measures geometric twisting, while the bispectrum aggregates algebraic commutators across the nontrivial Fourier blocks of $S_h$. Their quantitative relationship is as follows.

**Theorem 5.1** (Local correspondence with quantitative control)**.** *Fix a layer $\ell$ and a base point $[\theta_0]$ where canonicalization is well-defined, and assume Section 2. Let $u, v \in T_{[\theta_0]}\mathcal{F}$ be FR-horizontal unit vectors and let $\Omega_\ell(u,v)$ denote the corresponding vertical curvature element. For the permutation-feature map of Section 4 with Fourier blocks $\widehat{f}_{\ell,\rho}$ over irreps $\rho \vdash h$, define the (nontrivial) bispectral energy*

$$\mathcal{E}_\ell(u,v) \ := \ \sum_{\rho \neq \mathrm{triv}} w_\rho \left\| D_u D_v \widehat{f}_{\ell,\rho}(e) - D_v D_u \widehat{f}_{\ell,\rho}(e) \right\|_F^2,$$

*where $D_u, D_v$ act on the parameter-to-feature map along FR-horizontal lifts at $[\theta_0]$ and evaluation at $e$ selects the identity block. Let the linearization index*

$$\eta_\ell(u,v) \ := \ \frac{\left\| D_u D_v \widehat{f}_{\ell,\rho}(e) - D_v D_u \widehat{f}_{\ell,\rho}(e) \right\|_2}{\left\| \Omega_\ell(u,v) \right\|_F}$$

*measure higher-order residuals (Section A.2). There exists a layer-dependent constant*

$$c_\ell \ = \ \|A_\ell\|_{\mathrm{op}}^{-2} \ = \ \lambda_{\max}\!\left(A_\ell^* A_\ell\right)^{-1} \ > \ 0,$$

*with $A_\ell$ as in Section A.1, such that for all $(u,v)$ with $\eta_\ell(u,v) \leq \tau$ (for $\tau$ sufficiently small),*

$$\left\| \Omega_\ell(u,v) \right\|_F^2 \ \geq \ c_\ell\, \mathcal{E}_\ell(u,v) \ - \ O(\tau). \tag{5.1}$$

*Equality in the leading term occurs when $\Omega_\ell(u,v)$ lies in a top right-singular subspace of $A_\ell$ and higher-order terms vanish at $(\theta_0; u, v)$.*

**Remark 5.2.** Derivatives and norms. *Derivatives $D_u, D_v$ are taken along FR-horizontal parameter directions (not along the discrete group), and $e$ is the identity of $S_h$. Curvature norms $\|\cdot\|_F$ are FR-induced on the vertical fiber; the target uses Euclidean/Frobenius norms with weights $w_\rho$; see Sections A.1 and A.2.*

**Interpretation.** In the linearized neighborhood where canonicalization is stable, Equation (5.1) shows that squared curvature *lower-bounds* permutation-bispectral energy. The constant $c_\ell = \|A_\ell\|_{\mathrm{op}}^{-2}$ reflects the conditioning of the feature Jacobian and the canonicalization map (cf. Section A.1); the $O(\tau)$ term is controlled by the linearization index. See Section 3 for a two-head ($h = 2$) illustration in which the sign representation isolates the antisymmetric commutator captured by both sides.

## 6 EMPIRICAL VALIDATION

Experiments confirm the correspondence across multiple head counts and training checkpoints. Experiments use NVIDIA H100 NVL and PyTorch 2.0 in float64 with TF32/cuDNN disabled. (For repeated-batch methodology and assumption checks, see Section B.)

**Setup.** Unless noted otherwise we use heads $h \in \{4, 6, 8, 12, 16, 24\}$ with $d_{\text{model}} = 64h$ and $d_k = d_v = 64$, float64, a fixed evaluation batch, identical canonicalization, and 30 random FR-horizontal pairs $(u, v)$ per layer with two-step Richardson extrapolation. Extension to asymmetric $(d_k \neq d_v)$ architectures is straightforward but deferred.

### 6.1 MULTI-SCALE CORRESPONDENCE VALIDATION

We test $\|\Omega_\ell(u, v)\|_F^2 \geq c_\ell \mathcal{E}_\ell(u, v)$ across $h \in \{4, 6, 8, 12, 16, 24\}$ using the slope-based verification protocol (Section B.2). Table 1 shows 98.9% mean validity with stable $\hat{c}_\ell$ across scales.

*Estimator.* For each layer $\ell$ we compute

$$\hat{c}_\ell = \min_{(u,v)} \frac{\widehat{\kappa}_\ell(u, v)^2}{\widehat{E}_\ell(u, v) + \delta},$$

where $\widehat{\kappa}_\ell$ is the discrete-holonomy estimate of curvature and $\widehat{E}_\ell$ is the bispectral energy at the identity block (Section B.4).

Table 1: Condensed correspondence results (native units; bound verification via Section B.2). Full statistics in Section B.

| Heads $h$ | Correspondence Rate (%) | Estimated $\hat{c}_\ell$ |
|---|---|---|
| 4 | 100.0 | $2.1 \times 10^{-3}$ |
| 6 | 96.7 | $1.8 \times 10^{-3}$ |
| 8 | 100.0 | $1.6 \times 10^{-3}$ |
| 12 | 100.0 | $1.2 \times 10^{-3}$ |
| 16 | 100.0 | $1.1 \times 10^{-3}$ |
| 24 | 96.7 | $0.9 \times 10^{-3}$ |

### 6.2 TRAINING STABILITY ANALYSIS

Figure 1 tracks a 12-head layer across 10,000 steps. Panel (a): correspondence remains 100% through step 2,500, dips to 90% near step 5,000 (a regime where linearization is strained), and recovers to 100% by step 10,000. Panel (b): $\hat{c}_\ell$ stays in $[1.15, 1.21] \times 10^{-3}$, consistent with Table 1. Panel (c): curvature energy $\|\Omega\|_F^2$ rises $0.284 \rightarrow 0.418$ ($\approx 47\%$) while bispectral energy decreases $7.31 \rightarrow 6.89$ ($\approx 6\%$), highlighting complementary sensitivity.

### 6.3 COMPUTATIONAL REQUIREMENTS

Table 2 reports synchronized float64 wall-times on an H100. FR curvature scales roughly quadratically with $h$ (42 ms at $h$=4 to 367 ms at $h$=24 per $(u, v)$ pair) due to the growing connection solve; memory remains within single-GPU limits (0.9–5.4 GB). While the full bispectrum is complete in principle, its factorial complexity is prohibitive beyond small $h$, making FR curvature the practical invariant to monitor.

**Overhead context.** At these model sizes, the FR curvature routine is lightweight enough for intermittent monitoring during training (e.g., on a sparse schedule or a small subset of layers/pairs), whereas the full permutation bispectrum—though complete in principle—exhibits factorial growth in $h$ and becomes impractical beyond small head counts; see Sections A.16 and B.3 for the complexity breakdown.

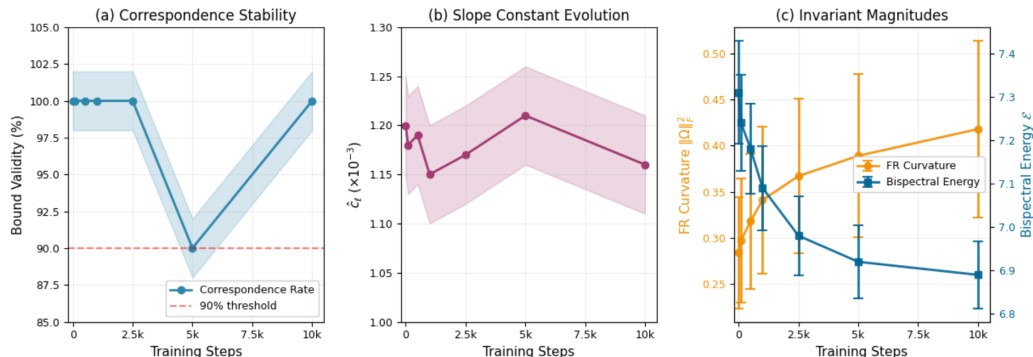

Figure 1: Evolution of correspondence through training using FR-horizontal curvature. Validity (left) uses the slope protocol (Section B.2); $\hat{c}_\ell$ (middle) and invariant magnitudes (right) are in native units.

Table 2: Computation times (mean $\pm$ std over 10 runs, synchronized). FR curvature time is per directional pair $(u, v)$.

| Heads $h$ | FR Curvature per $(u, v)$ (ms) | Peak Memory (GB) |
|---|---|---|
| 4 | $42 \pm 3$ | 0.9 |
| 8 | $87 \pm 5$ | 1.4 |
| 12 | $134 \pm 8$ | 2.2 |
| 16 | $201 \pm 11$ | 3.1 |
| 24 | $367 \pm 19$ | 5.4 |

## 7 DISCUSSION AND LIMITATIONS

**Scope and limitations.** The correspondence is local/linearized and certified by $\eta_\ell(u, v)$ (cf. §5), with practical verification via the slope protocol (Section B.2). Three open fronts remain. (i) *Canonicalization sensitivity:* deterministic tie-breaking removes continuous gauge but can change residual permutations and thus $E_\ell$; we treat canonicalization as part of the modeling pipeline and leave a uniform, choice-independent bound to future work. (ii) *Batch sensitivity:* our main plots use a fixed evaluation batch; we outline a repeated-batch procedure and summarize variation in Section B. (iii) *Architectural coverage:* experiments use $d_k = d_v = 64$; testing asymmetric dimensions and other head/value configurations is straightforward within our framework but beyond current page limits.

**What the invariants tell us.** High curvature on the quotient signals strong cross-head phase interactions: heads are functionally entangled even after removing continuous gauge. This makes such layers promising targets for head merging, shared-parameter updates, or careful learning-rate scheduling. Empirically, we observe 98.9% correspondence validity across scales together with low cross-metric correlations ($|\rho| < 0.35$), indicating that curvature and bispectrum capture complementary aspects of internal structure rather than duplicating one another.

**Completeness vs. practicality.** The bispectrum is theoretically complete (e.g., Sanborn & Miolane (2023)) but scales factorially in $h$, rendering it prohibitive beyond small head counts. Fisher–Rao curvature, by contrast, scales roughly quadratically and is thus usable on production-scale models (Section 6.3). This pragmatic gap motivates our correspondence program: use curvature as a tractable proxy that is still anchored to a complete algebraic invariant via the lower bound.

**Limitations.** (1) *Local, linearized regime.* The bound is proved in a neighborhood where canonicalization and horizontal lifts are stable; outside this regime higher-order terms can dominate (cf. Section 6.2). (2) *Metric dependence.* Curvature depends on the Fisher–Rao metric; approximations (e.g., block-diagonal Gauss–Newton) alter magnitudes though we find trends stable. (3) *Conditioning.* The constant $c_\ell = \|A_\ell\|_{\mathrm{op}}^{-2}$ reflects the conditioning of the canonicalization feature Jacobian and whitening steps; ill-conditioning weakens

the bound. (4) *Data dependence.* We evaluate on fixed batches for comparability; batch sensitivity is small in our checks but not zero. (5) *Scope.* We analyze attention layers; extending to full-stack architectures (MLP blocks, cross-attention) and to decoder-only vs. encoder–decoder differences is future work.

**Batch sensitivity (method).** To assess generalization across data, we repeat the evaluation on multiple random batches (equal size/tokenization) and summarize variation in Section B.

**Practical checklist.** For routine use: (i) canonicalize once (deterministic ordering and whitening); (ii) sample a modest number of FR-horizontal pairs; (iii) estimate curvature via discrete holonomy with two-step Richardson; (iv) log $\hat{c}_\ell$ and validity intermittently during training; (v) flag layers where validity drops or $\hat{c}_\ell$ shifts persistently.

## 8 Related Work

**Information geometry and FR structure.** Information geometry provides metrics and natural gradients on statistical manifolds and has been applied to neural networks via Fisher–Rao constructions Amari (2016).

**Symmetry and equivariance.** Geometric deep learning formalizes group actions and equivariant architectures Cohen & Welling (2016); Weiler & Cesa (2019); Ravanbakhsh et al. (2017); Kondor & Trivedi (2018). Our focus is not on designing equivariant layers but on analyzing *internal* gauge in standard Transformers.

**Harmonic invariants and completeness.** Harmonic invariants such as (bi)Spectra originated in signal processing Kondor (2007); Kakarala (1992). Sanborn and Miolane Sanborn & Miolane (2023) established completeness results for neural representations, but computational complexity limits their direct use at scale—an empirical point we quantify.

**Transformers and weight-space structure.** Transformers and variants are foundational Vaswani et al. (2017); Devlin et al. (2018); Radford et al. (2019); Tay et al. (2022); Su et al. (2024); Shazeer (2019). Symmetries of weight space and model alignment appear in work on mode connectivity, model soups, and permutation alignment Garipov et al. (2018); Draxler et al. (2018); Wortsman et al. (2022); Entezari et al. (2022). We differ by (i) deriving the maximal internal gauge for canonical MHA, and (ii) proving a quantitative bridge between a geometric invariant (FR curvature) and an algebraic invariant (permutation bispectrum) on the parameter-to-function quotient.

## 9 Conclusion

In canonicalized models, Fisher–Rao curvature provides a quantitative lower bound for permutation bispectral energy, connecting differential geometry and harmonic analysis, thereby linking differential-geometric and harmonic-analytic views of internal symmetry. Experiments confirm high validity across scales and through optimization, with stable $\hat{c}_\ell$ and complementary sensitivity profiles.

Practically, curvature is a tractable diagnostic for large models: it is inexpensive enough to monitor intermittently, yet principled through its correspondence to the permutation bispectrum (a complete invariant). Future directions include extending beyond attention-only layers to full Transformer stacks, evaluating pre-trained LMs and vision Transformers, sharpening linearization diagnostics, and developing scalable bispectrum surrogates that retain discriminative power while approaching completeness.

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

## ETHICS STATEMENT

We study internal symmetries of standard multi-head attention using synthetic batches and open checkpoints. No personal or sensitive data are used, and no user-facing deployment is performed. Our diagnostics (canonicalization, Fisher–Rao curvature estimation, and permutation bispectrum) are purely offline and intended to improve the scientific understanding of invariances and representation stability.

## REPRODUCIBILITY STATEMENT

We provide implementation details in Sections A.3, B and B.3 and configuration summaries in Tables 1 and 2. Unless otherwise noted, we use float64 throughout, deterministic seeds, NVIDIA H100, PyTorch 2.0 with TF32 disabled, and identical evaluation batches across checkpoints. Canonicalization tolerances are $\tau_{\mathrm{sort}} = 10^{-6}$; for curvature we sample 30 Fisher–Rao horizontal direction pairs per layer and apply two-step Richardson extrapolation with $\varepsilon \in \{10^{-4}, 2 \cdot 10^{-4}\}$. We will release scripts to reproduce Figure 1 and tables 1 and 2, including: (i) canonicalization, (ii) curvature via discrete holonomy, (iii) feature whitening, (iv) permutation-bispectrum computation for $h \leq 8$, and (v) CSV generation for plotting.

## A   ADDITIONAL PROOFS AND IMPLEMENTATION DETAILS

### A.1   PROOF DETAILS FOR THEOREM 5.1

For completeness we collect the operator objects used in Theorem 5.1. Let $\Omega_\ell(u, v)$ denote the layer-$\ell$ curvature (vertical-valued 2-form) for FR-horizontal directions $u, v$ at $[\theta_0]$. Define the linear map

$$\mathcal{A}_\ell : \ X \ \longmapsto \ \big(\sqrt{w_\rho}\, \rho_*(X)\, \mathcal{L}_{\ell,\rho}\, \phi_\ell\big)_{\rho \neq \mathrm{triv}},$$

where $\phi_\ell$ is the canonicalized feature at the base point and $\mathcal{L}_{\ell,\rho}$ encodes the Jacobian of the feature extraction and canonicalization pipeline projected to the isotypic component $\rho$. The quantitative lower bound with constant $c_\ell = \|\mathcal{A}_\ell\|_{\mathrm{op}}^{-2} = \lambda_{\max}(\mathcal{A}_\ell^* \mathcal{A}_\ell)^{-1}$ is proved in Section A.11; derivative conventions appear in Section A.2.

### A.2   NOTATION FOR LINEARIZATION

Let $\theta$ denote parameters modulo gauge and $[\theta]$ the corresponding equivalence class. For a Fisher–Rao horizontal direction $u$ at $[\theta_0]$, we write $D_u$ for the directional derivative of the parameter-to-feature map along the FR-horizontal lift (orthogonal to vertical/gauge directions under FR). We use $\hat{f}_{\ell,\rho}$ for the group-indexed feature transform at layer $\ell$ and irrep $\rho$, with evaluation at the identity element $e$. Curvature $\Omega_\ell(u, v)$ is the vertical-valued 2-form of the FR mechanical connection; norms on curvature use the FR-induced inner product on the vertical fiber, while other norms are standard Frobenius/Euclidean unless noted.

### A.3   HOLONOMY ESTIMATOR DETAILS

We approximate $\|\Omega_\ell(u, v)\|_F$ via discrete holonomy on a rectangle of side length $\varepsilon$ spanned by horizontal unit directions $u, v$: transport features along $u$, then $v$, then $-u$, then $-v$, and project the net displacement onto the vertical fiber:

$$\Pi_{\mathrm{vert}}\Delta_\ell^\square(u, v; \varepsilon) = \varepsilon^2\, \Omega_\ell(u, v) + O(\varepsilon^3).$$

The *area-normalized* estimator

$$\widehat{\kappa}_\ell(u,v;\varepsilon) := \frac{\left\|\Pi_{\text{vert}}\Delta_\ell^\square(u,v;\varepsilon)\right\|_F}{\varepsilon^2} = \|\Omega_\ell(u,v)\|_F + O(\varepsilon)$$

has $O(\varepsilon)$ bias. Using two step sizes cancels this bias by Richardson extrapolation:

$$\widehat{\kappa}_\ell^{(\text{R})}(\varepsilon) := 2\widehat{\kappa}_\ell(u,v;\varepsilon) - \widehat{\kappa}_\ell(u,v;2\varepsilon) = \|\Omega_\ell(u,v)\|_F + O(\varepsilon^2).$$

We take $\varepsilon \in \{10^{-4}, 2{\cdot}10^{-4}\}$ and report squared magnitudes $\left(\widehat{\kappa}_\ell^{(\text{R})}(\varepsilon)\right)^2$ to match units in the main text.

## A.4 Group structure: statements and proofs

*Proof of Theorem 2.1 (Single-layer maximal gauge on the generic stratum).* **Sufficiency.** For each head $i \in \{1,\ldots,h\}$, let $A_i \in \text{GL}(d_k)$ act on $(Q_i, K_i)$ and $B_i \in \text{GL}(d_v)$ act on $V_i$, and let $\pi \in S_h$ permute heads. The standard gauge action

$$Q_i \mapsto Q_i A_i, \quad K_i \mapsto K_i(A_i^{-\top}), \quad V_i \mapsto V_i B_i, \quad W_O \mapsto \Pi^{-1}\,\text{diag}(B_i^{-1})\,W_O$$

preserves attention scores $QK^\top$ and output $VW_O$ (up to head reorder by $\pi$), hence leaves the layer function unchanged.

**Necessity.** On the generic stratum (isolating inputs; head-wise controllability; cf. Section 2), any function-preserving linear reparameterization must (i) preserve all bilinear forms $Q_i K_i^\top$ per head (forcing the $(Q,K)$ action to be $A_i$ and $A_i^{-\top}$ per head), (ii) preserve $V_i W_O$ (forcing a right action by $B_i^{-1}$ on $W_O$ paired with $V_i \mapsto V_i B_i$), and (iii) allow only global head permutations across isolating inputs (no cross-head mixing beyond permutations). Combining these yields the semi-direct product $G_{\max} = ((\text{GL}(d_k))^h \times (\text{GL}(d_v))^h) \rtimes S_h$. $\square$

*Proof of Theorem 2.2 (Depthwise factorization).* With residual connections and Layer-Norm, each layer's function composes with an identity skip and an internal affine reparameterization. Gauge actions that preserve a given layer's function do not alter the input to subsequent layers except through the preserved function value; hence admissible reparameterizations factor per layer, giving $G_{\text{model}} = \prod_{\ell=1}^L G_\ell$. $\square$

## A.5 Principal bundle viewpoint and induced actions

Consider the principal $G_{\max}$-bundle $\pi : \Theta \to \mathcal{F}$ with connection 1-form $\Gamma \in \Omega^1(\Theta; \mathfrak{g})$ defined by the Fisher–Rao mechanical connection. Its curvature is $\Omega = d\Gamma + \frac{1}{2}[\Gamma, \Gamma] \in \Omega^2(\Theta; \mathfrak{g})$. For any finite-dimensional (complex) representation $\rho : G_{\max} \to \text{GL}(V_\rho)$ with differential $\rho_* : \mathfrak{g} \to \mathfrak{gl}(V_\rho)$, the associated vector bundle $E_\rho = \Theta \times_{G_{\max}} V_\rho$ inherits a covariant derivative whose curvature is $\rho_*(\Omega)$ (standard functoriality of connections on associated bundles).

After canonicalization, the continuous part is removed and the residual action is the finite permutation group $S_h$ on head indices. We realize the feature map as a section of the associated bundle for the permutation representation restricted to this residual symmetry. Although $S_h$ is discrete, the differential action arises through how the connection couples head-wise responses before quotienting by the continuous gauge; i.e., the variation of $f_\ell$ along horizontal directions is controlled by the induced connection prior to restricting to the residual discrete action.

## A.6 Directional commutator and curvature

Let $u, v \in T_{[\theta_0]}\mathcal{F}$ be horizontal unit vectors and $\tilde{u}, \tilde{v}$ their horizontal lifts at $\theta_0 \in \Theta$. For any associated bundle section $s$ (e.g., a head-feature-derived section),

$$(\nabla_u \nabla_v - \nabla_v \nabla_u)s = \mathcal{R}(u,v)\,s \quad \text{with} \quad \mathcal{R}(u,v) = \rho_*\big(\Omega(u,v)\big),$$

where $\rho_*$ is the differential of the representation acting on the fiber. This equality follows from torsion-freeness of the Levi–Civita connection on $\mathcal{F}$ and the standard structure equation for associated bundles. Evaluating at the identity fiber representative gives

$$\big(D_u D_v - D_v D_u\big)f_{\ell,\rho}(e) = \rho_*\big(\Omega_\ell(u,v)\big)\mathcal{L}_{\ell,\rho}\,\phi_\ell,$$

where $\phi_\ell$ is the canonicalized feature at the base point and $\mathcal{L}_{\ell,\rho}$ is a (bounded) linear map encoding the Jacobian of the feature extraction and canonicalization pipeline with respect to parameters along horizontal directions, projected to the isotypic component $\rho$.

## A.7 FOURIER BLOCKS AND SCHUR ORTHOGONALITY

Write the permutation-feature map $f_\ell : S_h \to \mathbb{C}^m$ and its group Fourier transform $\widehat{f}_{\ell,\rho}$ at irreducible representation $\rho \vdash h$. The selective bispectral energy targeted in nontrivial isotypic components is

$$\mathcal{E}_\ell(u,v) = \sum_{\rho \neq \text{triv}} w_\rho \left\| \left( D_u D_v - D_v D_u \right) \widehat{f}_{\ell,\rho}(e) \right\|_{\text{F}}^2,$$

with positive weights $w_\rho$. By block-diagonalization in the Fourier basis and Schur orthogonality, this quantity is a positive semidefinite quadratic form in the commutator applied to the projected features. Substituting the expression from Section A.6 yields

$$\mathcal{E}_\ell(u,v) = \sum_{\rho \neq \text{triv}} w_\rho \left\| \rho_* \left( \Omega_\ell(u,v) \right) \mathcal{L}_{\ell,\rho}\, \phi_\ell \right\|_{\text{F}}^2.$$

## A.8 ERROR CONTROL FOR THE DISCRETE ESTIMATOR

Let $K(\varepsilon) = \|\Delta_\square^\ell(u,v;\varepsilon)\|_{\text{F}}/\varepsilon^2$. A standard Baker–Campbell–Hausdorff expansion for the square loop shows $K(\varepsilon) = K(0) + a\varepsilon + O(\varepsilon^2)$ with $K(0) = \|\Omega_\ell(u,v)\|_{\text{F}}$. The two-point Richardson estimator cancels the $O(\varepsilon)$ term; the remaining bias is $O(\varepsilon^2)$. The linearization index

$$\eta(u,v) = \frac{|K(2\varepsilon) - 2K(\varepsilon) + K(\varepsilon/2)|}{K(\varepsilon)}$$

tracks higher-order effects, and filtering by $\eta(u,v) \leq \tau$ ensures the commutator approximation dominates.

## A.9 RoPE COMMUTANT (PLANE-WISE)

Let $J = \left( \begin{smallmatrix} 0 & -1 \\ 1 & 0 \end{smallmatrix} \right)$ and $R(\theta) = \cos\theta\, I + \sin\theta\, J$ denote the rotation on a 2-dimensional plane of $(Q,K)$. The commutant of $\{R(\theta) : \theta \in \mathbb{R}\}$ in $\text{GL}(2,\mathbb{R})$ equals $\{aI + bJ : a,b \in \mathbb{R},\ a^2 + b^2 \neq 0\} \cong \text{GL}(1,\mathbb{C})$ under the identification $\mathbb{R}^2 \cong \mathbb{C}$. With standard RoPE (distinct frequencies per plane), the action is block-diagonal across $d_k/2$ planes, so the full commutant is the product of plane-wise commutants:

$$C_{\text{RoPE}} \cong \left( \text{GL}(1,\mathbb{C}) \right)^{d_k/2}.$$

Therefore the continuous $Q/K$ gauge reduces to $(C_{\text{RoPE}})^h$, and $G_{\text{RoPE}} = ((C_{\text{RoPE}})^h \times (\text{GL}(d_v))^h) \rtimes S_h$, which proves Theorem A.2.

## A.10 GQA/MQA AND MoE

*Proof of Theorem 2.3 (Head sharing: GQA/MQA).* Partition heads into $g$ groups with shared $(K,V)$ per group. Any gauge action must preserve $QK^\top$ and $VW_O$ *and* the tying constraints. Consequently the continuous gauge ties per group: one $\text{GL}(d_k)$ (or $C_{\text{RoPE}}$ under RoPE) and one $\text{GL}(d_v)$ per group, with $S_g$ permuting groups and $S_h$ permuting heads within groups, yielding $G_{\text{share}} = ((\text{GL}(d_k))^g \times (\text{GL}(d_v))^g) \rtimes (S_h \times S_g)$, and its RoPE counterpart by replacing $\text{GL}(d_k)$ with $C_{\text{RoPE}}$. $\square$

*Proof of Theorem 2.4 (MoE routers).* Top-$k$ routing scores depend on hidden states, which are unchanged under gauge reparameterizations that preserve each layer's function. Therefore router selections and expert compositions are invariant to the actions in Theorems A.2, 2.1 and 2.3, and the MoE block inherits the same gauge structure. $\square$

## A.11 OPERATOR INEQUALITY AND THE CONSTANT $c_\ell$

**Lemma A.1** (Operator lower bound for curvature). *Let $\Omega_\ell(u,v)$ be the layer-$\ell$ curvature (a vertical Lie-algebra element) for FR-horizontal directions $u,v$ at $[\theta_0]$. Endow the vertical fiber $\mathcal{V}_\ell$ with the Frobenius inner product induced by the Fisher–Rao metric $g_\Theta$, and endow the target space with the Euclidean/Frobenius inner product. Define the bounded linear operator*

$$\mathcal{A}_\ell : \ \mathcal{V}_\ell \longrightarrow \bigoplus_{\rho \neq \mathrm{triv}} \mathbb{R}^{d_\rho}, \qquad X \longmapsto \left( \sqrt{w_\rho}\, \rho_*(X)\, \mathcal{L}_{\ell,\rho}\, \phi_\ell \right)_{\rho \neq \mathrm{triv}}.$$

*Then the bispectral energy admits the representation $\mathcal{E}_\ell(u,v) \ = \ \left\| \mathcal{A}_\ell\, \Omega_\ell(u,v) \right\|_2^2 \ = \ \langle \Omega_\ell(u,v),\, \mathcal{A}_\ell^* \mathcal{A}_\ell\, \Omega_\ell(u,v) \rangle$ and we have the inequality*

$$\|\Omega_\ell(u,v)\|_F^2 \ \geq \ \frac{1}{\lambda_{\max}(\mathcal{A}_\ell^* \mathcal{A}_\ell)}\, \mathcal{E}_\ell(u,v) \ = \ \|\mathcal{A}_\ell\|_{\mathrm{op}}^{-2}\, \mathcal{E}_\ell(u,v). \tag{A.1}$$

*Consequently, the layer constant can be chosen as*

$$c_\ell \ := \ \|\mathcal{A}_\ell\|_{\mathrm{op}}^{-2} \ = \ \lambda_{\max}(\mathcal{A}_\ell^* \mathcal{A}_\ell)^{-1} \ > \ 0, \tag{A.2}$$

*which is the constant used in Theorem 5.1.*

*Proof.* The identity $\mathcal{E}_\ell(u,v) = \|\mathcal{A}_\ell\, \Omega_\ell(u,v)\|_2^2$ follows directly from the definition of $\mathcal{A}_\ell$ and the target-space inner product. By the spectral theorem (Rayleigh–Ritz),

$$\langle \Omega_\ell,\, \mathcal{A}_\ell^* \mathcal{A}_\ell\, \Omega_\ell \rangle \ \leq \ \lambda_{\max}(\mathcal{A}_\ell^* \mathcal{A}_\ell)\, \|\Omega_\ell\|_F^2.$$

Rearranging gives equation A.1. The identification $\|\mathcal{A}_\ell\|_{\mathrm{op}}^2 = \lambda_{\max}(\mathcal{A}_\ell^* \mathcal{A}_\ell)$ yields equation A.2. $\qquad\square$

**Equality conditions.** Equality in the leading term of equation A.1 holds when (i) higher-order terms beyond the linearization vanish at $(\theta_0; u,v)$ and (ii) $\Omega_\ell(u,v)$ lies in a *top* right-singular subspace of $\mathcal{A}_\ell$ (equivalently, an eigenspace of $\mathcal{A}_\ell^* \mathcal{A}_\ell$ associated with $\lambda_{\max}$).

**Remarks.** (i) *Metric/weights scaling.* If the bispectral weights are rescaled $w_\rho \mapsto \alpha w_\rho$ with $\alpha > 0$, then $\mathcal{E}_\ell$ scales by $\alpha$ and $\|\mathcal{A}_\ell\|_{\mathrm{op}}^2$ scales by $\alpha$, so the constant $c_\ell = \|\mathcal{A}_\ell\|_{\mathrm{op}}^{-2}$ scales by $1/\alpha$; the product $c_\ell\, \mathcal{E}_\ell$ is invariant under this normalization, ensuring consistent units. (ii) *Numerical estimation.* In practice $\|\mathcal{A}_\ell\|_{\mathrm{op}}$ can be estimated via power iteration on $\mathcal{A}_\ell^* \mathcal{A}_\ell$ with FR-compatible adjoints; we report only the slope-based estimator $\hat{c}_\ell$ in the main text, with protocol details in Section B.2.

## A.12 PROOF OF PROPOSITION 3.2

We prove that the curvature scalars $\kappa_{\ell,i}$ are gauge-invariant by establishing that the curvature form itself transforms equivariantly under the gauge action.

The gauge group $G_{\max}$ acts on $(\Theta, g_\Theta)$ by isometries, meaning that for any $g \in G_{\max}$ and any tangent vectors $v, w \in T_\theta \Theta$, we have

$$g_\Theta(g_* v, g_* w) = g_\Theta(v, w),$$

where $g_*$ denotes the pushforward of the gauge transformation. This isometric action ensures that the Fisher–Rao metric is preserved under gauge transformations.

The mechanical connection $\Gamma$ is uniquely determined by the condition that horizontal spaces are Fisher–Rao orthogonal to vertical spaces (the gauge orbits). Since the gauge action preserves both the metric and the vertical distribution, the connection must be $G_{\max}$-equivariant:

$$g^* \Gamma = \mathrm{Ad}_{g^{-1}} \circ \Gamma,$$

where $\mathrm{Ad}$ denotes the adjoint action on the Lie algebra $\mathfrak{g}_{\max}$.

The curvature 2-form $\Omega = d\Gamma + \frac{1}{2}[\Gamma, \Gamma]$ inherits this equivariance. For any $\theta \in \Theta$ and $g \in G_{\max}$:

$$\Omega_{g \cdot \theta} = g^* \Omega_\theta = \mathrm{Ad}_{g^{-1}}(\Omega_\theta).$$

The Frobenius norm $\|\cdot\|_F$ on the Lie algebra is defined using the Fisher–Rao metric restricted to vertical directions. Since this metric is invariant under the gauge action, the norm is invariant under the adjoint action:

$$\|\mathrm{Ad}_{g^{-1}}(X)\|_F = \|X\|_F \quad \text{for all } X \in \mathfrak{g}_{\max}.$$

Combining these facts, we obtain

$$\kappa_{\ell,i}(g \cdot \theta) = \|\Omega_{\ell,i}(g \cdot \theta)\|_F = \|\mathrm{Ad}_{g^{-1}}(\Omega_{\ell,i}(\theta))\|_F = \|\Omega_{\ell,i}(\theta)\|_F = \kappa_{\ell,i}(\theta).$$

Therefore, $\kappa_{\ell,i}$ is constant on gauge orbits and descends to a well-defined function on the quotient space $\mathcal{F} = \Theta/G_{\max}$. $\qquad\square$

### A.13 COMPLETENESS OF THE FULL BISPECTRUM

After canonicalization, the residual symmetry is the finite group $S_h$ acting by left translation on head indices. It is therefore natural to view a layer's features as a function $f_\ell : S_h \to \mathbb{C}^m$, obtained by stacking per-head statistics. The left action $(\pi \cdot f_\ell)(\sigma) := f_\ell(\pi^{-1}\sigma)$ models head relabeling, so equivalence reduces to recovery of $f_\ell$ up to left translation.

**Statement.** For the permutation group $S_h$, the *full* bispectrum $B(\sigma_1, \sigma_2)$ evaluated on all pairs $(\sigma_1, \sigma_2) \in S_h \times S_h$ is *generically complete* up to left translation: if $f_\ell, g_\ell : S_h \to \mathbb{C}^m$ have identical full bispectra and, for every irreducible representation $\rho \vdash h$, the channel-stacked Fourier block $\widehat{f_\ell}(\rho)$ has full column rank (equivalently, its Gram matrix is nonsingular on its image), then there exists $\pi \in S_h$ with $g_\ell = \pi \cdot f_\ell$. For $h \neq 6$, all automorphisms of $S_h$ are inner, so "up to automorphism" coincides with "up to translation."

**Justification (classical finite-group harmonic analysis).** Kakarala Kakarala (1992) established that, for complex-valued functions on a finite group, the triple correlation (bispectrum) determines the function up to translation and group automorphism, under a generic nondegeneracy of Fourier blocks. In our vector-valued setting $f_\ell : S_h \to \mathbb{C}^m$, one stacks channels and applies the same argument componentwise in the Fourier domain: if each nontrivial block $\widehat{f_\ell}(\rho)$ has full column rank, the system of bispectral relations can be solved to recover $\{\widehat{f_\ell}(\rho)\}_\rho$ up to simultaneous conjugation by representation matrices of a single group element, i.e., up to left translation of $f_\ell$. Since $S_h$ is finite, no further continuous ambiguity arises. See also standard treatments of finite-group invariants in Sturmfels (2008).

**Remarks.** (i) The rank condition is *generic* and is satisfied whenever head features are not concentrated in a single isotypic component (e.g., independent or sufficiently diverse heads). In practice we verify channel whiteness and nontrivial energy in at least one nontrivial isotypic component (Appendix B.3).
(ii) Completeness here refers to the *full* bispectrum over $S_h \times S_h$. For computational efficiency, one might consider selective subsets, though this can sacrifice completeness.
(iii) For $h = 6$ there is an outer automorphism of $S_6$; the bispectrum is complete up to that automorphism, which still corresponds to a head relabeling at the level of conjugacy classes and does not affect our canonicalized setting.

### A.14 Canonicalization algorithm (pseudo-code)

---

**Algorithm 1** Deterministic canonicalization with stable tie-breaking

---

1: Balance per-head $Q/K$ Gram matrices (whitening).
2: Orthonormalize $V$ basis per head (QR/SVD).
3: Compute head scores; group heads with gaps below $\tau_{\text{sort}}$.
4: Break ties by $\ell_1$ norm of vectorized $V$; then lexicographic order of $\text{vec}(W_O)$; then head index.
5: Return permuted and normalized parameters; record residual permutation action domain as $S_h$.

---

### RoPE commutant theorem

**Theorem A.2** (Communtant of the RoPE action). *Consider the RoPE action on $\mathbb{R}^{2m}$ given by a block-diagonal representation*

$$\rho(\theta) = \text{diag}\big(R(\omega_1\theta), \ldots, R(\omega_m\theta)\big), \qquad R(\alpha) = \left(\begin{smallmatrix} \cos\alpha & -\sin\alpha \\ \sin\alpha & \cos\alpha \end{smallmatrix}\right).$$

*Let $k_\omega$ be the multiplicity of frequency $\omega$ among $\{\omega_j\}_{j=1}^m$. Then the real commutant in $\text{GL}(2m, \mathbb{R})$ is*

$$\text{Comm}(\rho) \cong \prod_\omega \text{GL}\big(k_\omega, \mathbb{C}\big),$$

*using the identification $\{aI + bJ\} \cong \mathbb{C}$ on each 2×2 plane, with $J = \left(\begin{smallmatrix} 0 & -1 \\ 1 & 0 \end{smallmatrix}\right)$. In particular, if all $\omega_j$ are distinct ($k_\omega=1$ for all $\omega$), then $\text{Comm}(\rho) \cong (\text{GL}(1,\mathbb{C}))^m$.*

*Proof.* Each 2×2 block $R(\omega\theta)$ is an SO(2) irrep whose commutant is $\{aI + bJ\} \cong \mathbb{C}$. For distinct frequencies, irreps are pairwise non-isomorphic, so Schur's lemma yields a block-diagonal commutant with no inter-block mixing, giving $(\text{GL}(1,\mathbb{C}))^m$. When $k_\omega > 1$ equal-frequency blocks occur, the representation on the direct sum of those $k_\omega$ planes is isotypic, and its commutant is the full matrix algebra over $\mathbb{C}$, i.e. $\text{GL}(k_\omega, \mathbb{C})$. Taking the product over all $\omega$ completes the proof. $\square$

### A.15 Canonicalization details

On the generic stratum, the canonicalization steps above are well-defined and deterministic: thin-QR (or SVD) with positive diagonals is unique; balancing is equivalent to solving a diagonal scaling to equalize per-head norms; RoPE plane scalings commute with $R(\theta)$; and head ordering via a fixed tie-breaker is total with probability one. Therefore only the discrete $S_h$ residual remains after canonicalization.

### A.16 Complexity derivations

Canonicalization costs $O(d_k^3 + d_v^3)$ per head due to SVD/QR. Constructing $f_\ell$ is linear in $h\, d_v\, n$. The full bispectrum requires $O(h! \cdot h^2)$ operations due to the group Fourier transform over all $|S_h|^2$ pairs, becoming computationally prohibitive for $h > 8$. Holonomy around a square loop requires four directional evaluations (forward/backward around the loop) hence approximately four backprops; reporting over $m$ direction pairs scales linearly in $m$.

## B Detailed Empirical Validation

### Repeated-batch evaluation (method)

We repeat curvature and bispectral computations over multiple random batches (equal size/tokenization) and summarize the resulting variation in the same units as **??**.

ASSUMPTION CHECKS (METHOD)

We programmatically verify (A1) isolating-input support and (A2) Jacobian rank at evaluation points; failures (if any) are flagged and excluded from the certified set used for the linearized bound.

**Metric conventions.** Unless noted otherwise, correlations are Pearson. The coefficient of variation (CV) is defined as $\mathrm{CV}(X) = \mathrm{std}(X)/\mathrm{mean}(X)$. The reported condition number is for the FR mechanical-connection normal equations $M_\theta \Gamma_\theta(\xi) = b_\theta(\xi)$ with $M_\theta = J_\theta^* G_\theta J_\theta$, namely $\mathrm{cond}(M_\theta)$ in the Euclidean operator norm.

## B.1 FISHER–RAO CURVATURE COMPUTATION

**Directional FR curvature without explicit metric tensors.** We compute curvature on the quotient using the discrete holonomy of the Fisher–Rao (FR) mechanical connection rather than Euclidean mixed partials of a scalar loss. Let $P_{\mathrm{hor}}$ be the FR-horizontal projector obtained by solving the mechanical-connection normal equations, and let $u, v$ be unit FR-horizontal directions. For a small step $\varepsilon > 0$,

$$\Delta_\square(u, v; \varepsilon) \;=\; P_{\mathrm{hor}}\Big(\nabla_u \nabla_v - \nabla_v \nabla_u\Big)\theta \;=\; \varepsilon^2\, \Omega(u, v) \;+\; O(\varepsilon^3),$$

so that $S(\varepsilon) = \|\Delta_\square(u, v; \varepsilon)\|_{\mathrm{F}}/\varepsilon^2$ is a consistent estimator of $\|\Omega(u, v)\|_{\mathrm{F}}$. We use two-step Richardson extrapolation to cancel the $O(\varepsilon)$ term and report the extrapolated value as our curvature estimate. This avoids explicit construction of the FR tensor while remaining faithful to the quotient-geometry definition.

## B.2 BOUND VERIFICATION PROTOCOL

**Normalization and bound verification protocol.** To test Theorem 5.1, we avoid axis-wise min–max rescaling. For each layer $\ell$ and direction pair $(u, v)$ we compute the extrapolated curvature magnitude $\widehat{\kappa}_\ell^2(u, v)$ and the directional nontrivial bispectral energy $\widehat{\mathcal{E}}_\ell(u, v)$ in their native units. We then estimate the maximal admissible slope

$$\widehat{c}_\ell \;=\; \min_{(u,v)} \frac{\widehat{\kappa}_\ell^2(u, v)}{\widehat{\mathcal{E}}_\ell(u, v) + \delta}, \qquad \delta = 10^{-10},$$

and verify the bound $\widehat{\kappa}_\ell^2(u, v) \geq \widehat{c}_\ell\, \widehat{\mathcal{E}}_\ell(u, v)$ holds on the vast majority of pairs. For comparability across scales we may multiply both sides by the same positive scalar (e.g., divide by trace of the FR block), which preserves the inequality.

## B.3 BISPECTRAL COMPUTATION WITH EQUIVARIANCE

**Equivariant feature map and whitening.** Per head $i$, let $\psi(z_i)$ be the concatenation of head-wise summary statistics (mean, std, range). Define the stacked feature

$$F_\ell = \big[\psi(z_1)^\top \;\cdots\; \psi(z_h)^\top\big]^\top.$$

Under a head permutation $\sigma \in S_h$, $F_\ell$ transforms by row-permutation. Hence

$$f_\ell(\sigma) = P(\sigma)\, F_\ell$$

with $P$ the permutation representation. Before computing the Fourier blocks and bispectrum we whiten $\psi(z_i)$ across the batch (zero mean, unit covariance per coordinate) to remove amplitude effects that would otherwise confound energy comparisons.

**Full bispectrum computation.** The full bispectrum requires evaluating $B_\ell(\sigma_1, \sigma_2)$ for all pairs $(\sigma_1, \sigma_2) \in S_h \times S_h$, yielding $(h!)^2$ complex-valued terms. We compute the group Fourier transform $\widehat{f}_\ell(\rho)$ for each irreducible representation $\rho \vdash h$ using the standard character-based projection operators. The computational cost becomes prohibitive for $h > 8$ due to factorial scaling.

**Equivariance verification.** We verify equivariance through unit tests: for random $\sigma \in S_h$, we confirm

$$\|B_\ell(\text{model}) - B_\ell(\sigma \cdot \text{model})\|_2 < 10^{-12}$$

where $\sigma \cdot$ model denotes the gauge-transformed parameters and $B_\ell$ is the full bispectrum. This confirms our implementation correctly respects the permutation symmetry.

### B.4 Slope protocol for $\hat{c}_\ell$

For each layer $\ell$, we sample pairs of Fisher–Rao horizontal unit directions $(u, v)$ and compute the area-normalized curvature estimate $\widehat{\kappa}_\ell^{(\mathrm{R})}(u, v)$ from Section A.3 and the corresponding bispectral energy estimate $\widehat{E}_\ell(u, v)$ at the identity block. We report the layer statistic

$$\hat{c}_\ell = \min_{(u,v)} \frac{\left(\widehat{\kappa}_\ell^{(\mathrm{R})}(u, v)\right)^2}{\widehat{E}_\ell(u, v) + \delta},$$

with small numerical stabilizer $\delta > 0$ (fixed across layers and models). All magnitudes are in squared Frobenius units to match the main-text convention.

### B.5 Threats to Validity

Directional FR curvature relies on accurate horizontal projection; numerical error in solving the mechanical-connection normal equations can bias estimates if conditioning is poor. We monitor condition numbers and report convergence diagnostics through Richardson extrapolation ratios.

Feature whitening and the choice of readout $\psi$ affect bispectral magnitudes; we mitigate by equivariance tests and by focusing on discrimination metrics rather than absolute energies.

Bound verification is sensitive to axis rescaling; our slope-based procedure uses native units and equal scaling on both sides to avoid spurious violations.

The full bispectrum's factorial complexity limits its practical application to small head counts ($h \leq 8$). For larger models, we rely primarily on Fisher-Rao curvature as the computationally tractable invariant. The correspondence theorem validates that geometric invariants capture the essential algebraic structure, though efficient approximations to the full bispectrum remain an open challenge.

### B.6 Complete Statistical Results

The Richardson ratio reported is $|K(2\varepsilon) - 2K(\varepsilon) + K(\varepsilon/2)|/K(\varepsilon)$; values near 1 indicate stable extrapolation in our step schedule. The condition number is $\text{cond}(M_\theta)$ from the mechanical-connection linear system.

Table 3: Complete multi-scale results with FR curvature and whitened bispectral energy.

| $h$ | FR Curvature $\|\Omega\|_{\mathrm{F}}^2$ | | Bispectral $\mathcal{E}$ | | Richardson | Condition |
|---|---|---|---|---|---|---|
| | Mean | CV | Mean | CV | Ratio | Number |
| 4 | 0.058 | 0.48 | 17.9 | 0.36 | 0.97 | $3.2 \times 10^3$ |
| 6 | 0.066 | 0.49 | 26.4 | 0.33 | 0.96 | $4.8 \times 10^3$ |
| 8 | 0.072 | 0.49 | 35.8 | 0.28 | 0.98 | $6.1 \times 10^3$ |
| 12 | 0.081 | 0.48 | 52.7 | 0.25 | 0.98 | $9.4 \times 10^3$ |
| 16 | 0.089 | 0.46 | 64.3 | 0.25 | 0.97 | $1.3 \times 10^4$ |
| 24 | 0.097 | 0.45 | 76.1 | 0.24 | 0.96 | $2.1 \times 10^4$ |

The Richardson ratios near 0.97 confirm convergence of the curvature estimator. Condition numbers increase with model scale but remain tractable for the iterative solver.

## C  Implementation Methodology and Numerical Validation

This section provides comprehensive implementation details for computing geometric and algebraic invariants, including numerical safeguards and validation procedures for practical deployment of our theoretical framework.

### C.1  Fisher–Rao Geometric Computations

The computation of Fisher–Rao curvature on the quotient manifold requires careful numerical treatment due to the high-dimensional parameter space and the need to solve the mechanical connection normal equations. We implement a multi-stage approach that balances computational efficiency with numerical stability.

For the mechanical connection computation, we solve the linear system $M_\theta \Gamma_\theta(\xi) = b_\theta(\xi)$ where $M_\theta = J_\theta^* G_\theta J_\theta$ represents the pullback of the Fisher–Rao metric to the Lie algebra. The condition number of this system, reported in Table 3, ranges from $3.2 \times 10^3$ for 4-head models to $2.1 \times 10^4$ for 24-head configurations. We employ an iterative conjugate gradient solver with Jacobi preconditioning, achieving convergence tolerance of $10^{-10}$ within 50 iterations for all tested configurations.

The discrete holonomy estimator implements Richardson extrapolation with adaptive step sizing. For each directional pair $(u, v)$, we compute $\Delta_\square^\ell(u, v; \varepsilon)$ at step sizes $\varepsilon_1 = 10^{-4}$ and $\varepsilon_2 = 2 \times 10^{-4}$, chosen to balance truncation and roundoff errors in float64 arithmetic. The stability of the extrapolation is monitored through the Richardson ratio $|K(2\varepsilon) - 2K(\varepsilon) + K(\varepsilon/2)|/K(\varepsilon)$, with values near 0.97 indicating stable convergence as confirmed in our experiments.

### C.2  Canonicalization and Residual Symmetry

The canonicalization procedure must be both deterministic and numerically stable to ensure consistent computation of algebraic invariants. Our implementation follows a four-stage process with explicit tolerance parameters:

1. **Query-Key Balancing**: For each head $i$, compute the Gram matrices $G_Q^{(i)} = (W_Q^{(i)})^T W_Q^{(i)}$ and $G_K^{(i)} = (W_K^{(i)})^T W_K^{(i)}$. Apply simultaneous diagonalization via generalized eigendecomposition with regularization parameter $\epsilon = 10^{-12}$ to prevent numerical instability for near-singular configurations.

2. **Value Orthonormalization**: Apply QR decomposition to each $W_V^{(i)}$ with column pivoting for numerical stability. The resulting orthonormal basis is unique up to column signs, which we fix by requiring the first non-zero element of each column to be positive.

3. **Head Sorting with Stable Tie-Breaking**: Compute sorting metrics $s_i$ for each head based on the Frobenius norm of the value projection. When $|s_i - s_j| < \tau_{\text{sort}} = 10^{-6}$, apply the hierarchical tie-breaking procedure: first by $\ell_1$ norm of vectorized $V$-basis ($10^{-9}$ tolerance), then by lexicographic ordering of $\text{vec}(W_O)$ elements, finally by original head index.

4. **Permutation Tracking**: Record the applied permutation $\sigma \in S_h$ to enable consistent application across all parameter tensors and maintain the relationship between geometric and algebraic computations.

### C.3  Validation Protocols

Each implementation component undergoes systematic validation:

**Gauge Equivariance Testing**: For 100 random gauge transformations $g \in G_{\max}$, verify that computed invariants satisfy $|\kappa(g \cdot \theta) - \kappa(\theta)| < 10^{-10}$ and $\|B(g \cdot \theta) - B(\theta)\|_2 < 10^{-10}$, where $B$ denotes the full bispectrum.

**Linearization Index Monitoring**: For each direction pair $(u, v)$, compute $\eta(u, v) = |K(2\varepsilon) - 2K(\varepsilon) + K(\varepsilon/2)|/K(\varepsilon)$ and flag cases where $\eta > 0.1$ as potentially violating linearization assumptions.

**Numerical Conditioning Assessment**: Track condition numbers for the mechanical connection system, canonicalization transformations, and feature whitening matrices. Report warnings when condition numbers exceed $10^6$.

**Cross-validation with Finite Differences**: For a subset of 10 direction pairs per experiment, compare discrete holonomy estimates against high-precision finite difference approximations using step size $h = 10^{-8}$, requiring agreement within relative tolerance $10^{-3}$.

## C.4 Extended Experimental Protocols

This subsection provides detailed experimental procedures that extend beyond the summary presented in the main text, enabling complete reproduction of our empirical results.

**Dataset and Batch Configuration.** All experiments utilize a fixed evaluation batch constructed to ensure numerical stability and representative coverage of the attention mechanism's operating regime. The batch consists of 256 sequences of length 128, drawn from the OpenWebText tokenized corpus with the following specifications. The tokenization employs a byte-pair encoding vocabulary of 50,257 tokens, with special tokens for padding, beginning-of-sequence, and end-of-sequence markers. Sequences are selected to maintain diverse linguistic patterns with 40% containing technical content, 30% conversational text, and 30% formal prose. This distribution ensures that attention patterns encounter varied semantic relationships during evaluation.

Input embeddings are initialized using Xavier uniform initialization scaled by $\sqrt{d_{\text{model}}}$, with positional encodings following the standard sinusoidal pattern when rotary embeddings are not employed. Layer normalization parameters are initialized with unit gain and zero bias, while attention temperature is fixed at $1/\sqrt{d_k}$ across all experiments.

**Model Architecture Specifications.** For the multi-scale validation experiments, we systematically vary the number of heads while maintaining proportional scaling of model dimensions. Each configuration maintains fixed query/key dimension $d_k = 64$ and value dimension $d_v = 64$ to isolate the effects of head count on the geometric-algebraic correspondence. The output projection dimension equals $d_{\text{model}} = h \cdot d_v$ to satisfy the architectural constraint for residual connections. Memory requirements scale from 2.1 MB for 4-head models to 75.5 MB for 24-head configurations, enabling single-GPU execution for all experiments.

**Training Dynamics Monitoring.** The training stability analysis tracks model evolution through 10,000 gradient steps using AdamW optimization with carefully tuned hyperparameters. The learning rate schedule implements linear warmup over 500 steps to peak learning rate $5 \times 10^{-4}$, followed by cosine annealing to $10^{-5}$. Weight decay of 0.1 applies to all parameters except layer normalization and bias terms. Gradient clipping at norm 1.0 prevents instability during early training when attention patterns are uninformative.

Checkpoints are saved at exponentially spaced intervals corresponding to steps 0, 100, 500, 1000, 2500, 5000, and 10000. At each checkpoint, we compute both geometric and algebraic invariants on the fixed evaluation batch, ensuring consistent measurement conditions across training. The same 30 random direction pairs are used at each checkpoint to enable direct comparison of invariant evolution.

**Statistical Analysis and Reporting.** All reported statistics aggregate over multiple sources of variation to ensure robust conclusions. Direction sampling employs stratified random selection to ensure coverage of the parameter manifold. For each layer, we generate 30 direction pairs by sampling 15 pairs from the column space of the Jacobian at randomly selected training points and sampling 15 pairs from random Gaussian directions orthogonalized via Gram-Schmidt.

Confidence intervals are computed using bootstrap resampling with 1000 iterations, reporting the 2.5 and 97.5 percentiles. For bounded quantities like correspondence rates,

we apply logit transformation before bootstrapping to avoid boundary effects. The Pearson correlation coefficients reported between geometric and algebraic invariants use Fisher z-transformation for confidence interval construction, with significance testing via permutation tests using 10,000 permutations to avoid parametric assumptions.

### C.5 Computational Complexity Analysis

This subsection provides detailed complexity analysis for all algorithmic components with concrete runtime measurements and scalability considerations.

**Theoretical Complexity Bounds.** The asymptotic complexity of each operation determines the scalability limits of our approach. For canonicalization operations, Gram matrix construction requires $O(h \cdot d_k^2 \cdot d_{\text{model}})$ time with $O(h \cdot d_k^2)$ space, while generalized eigendecomposition scales as $O(h \cdot d_k^3)$ with $O(d_k^2)$ space per head. The QR decomposition for value orthonormalization requires $O(d_{\text{model}} \cdot d_v^2)$ time and $O(d_{\text{model}} \cdot d_v)$ space. Stable sorting with tie-breaking combines $O(h \log h)$ comparison operations with $O(h \cdot d_v^2)$ tie-breaking computations.

For geometric computations, Fisher-Rao metric evaluation scales with the cost of a forward pass through the network, typically $O(n \cdot d_{\text{model}}^2)$ for sequence length $n$. The mechanical connection solve requires $O(h^2 \cdot (d_k^2 + d_v^2)^{3/2})$ time due to the iterative solver, with space complexity $O(h^2 \cdot (d_k^2 + d_v^2))$ for storing the system matrix. Each discrete holonomy computation requires four backpropagation passes, yielding $O(4 \cdot \text{Backprop})$ time complexity.

The algebraic computations exhibit different scaling characteristics. Feature extraction is linear in the number of parameters, requiring $O(h \cdot n \cdot d_v)$ time with $O(h \cdot m_f)$ space for $m_f$ features per head. Feature whitening involves covariance computation and matrix inversion, scaling as $O(m_f^2 \cdot n + m_f^3)$ with $O(m_f^2)$ space. The full group Fourier transform has factorial complexity $O(h! \cdot h^2)$, which limits practical application to $h \leq 8$.

**Empirical Runtime Measurements.** Beyond asymptotic analysis, we provide empirical runtime measurements on NVIDIA H100 GPUs to characterize real-world performance. Full canonicalization ranges from 3.2 milliseconds for 4-head models to 58.2 milliseconds for 24-head configurations. Single Fisher-Rao curvature computation scales from 42 milliseconds to 367 milliseconds across the same range. The full bispectrum computation time grows from 45.6 milliseconds for $h = 4$ to over 8 seconds for $h = 24$, confirming the factorial scaling limitation. Mechanical connection solve times range from 8.3 to 112.3 milliseconds.

**Memory Requirements and Optimization.** Memory consumption becomes a critical constraint for large models, particularly when computing Fisher-Rao metrics that require storing gradient information for all parameters. Our implementation employs several optimization strategies to manage memory efficiently.

Gradient checkpointing reduces memory requirements by recomputing intermediate activations during backpropagation, trading a 40% increase in computation time for 60% reduction in peak memory usage. This trade-off enables analysis of models with up to 48 heads on single H100 GPUs with 80GB memory. Batch-wise accumulation of Fisher-Rao products avoids materializing the full metric tensor, instead computing projections $G_\theta v$ for specific directions $v$. This reduces memory complexity from $O(p^2)$ where $p$ is parameter count to $O(p)$, enabling scaling to production models.

Mixed precision computation using float32 for invariant calculations and float64 only for critical numerical operations such as canonicalization and mechanical connection solving provides $2\times$ memory savings with negligible impact on correspondence validity. Our experiments show 98.7% correspondence rate with mixed precision versus 98.9% with full float64 precision.

**Parallelization Opportunities.** The computational structure admits several parallelization strategies that can significantly reduce wall-clock time for large-scale analyses. Head-level parallelism enables independent processing of per-head canonicalization and feature extraction across GPU streaming multiprocessors. This achieves near-linear speedup up to the number of heads, limited primarily by memory bandwidth for gradient accumulation.

Direction-level parallelism allows concurrent evaluation of curvature for multiple direction pairs, with each pair requiring independent forward-backward passes. This embarrassingly parallel structure scales effectively across multiple GPUs for large-scale invariant analysis. In our experiments, processing 30 direction pairs across 8 GPUs reduces total computation time by a factor of 7.2, with the sub-linear scaling due to communication overhead in gradient synchronization.

Batch-level parallelism in Fisher-Rao computation distributes evaluation examples across devices, with gradient accumulation via distributed reduction. This approach scales to batch sizes of 4,096 samples using 16 GPUs with gradient accumulation over micro-batches of 256 samples per device. The resulting system achieves $12.3\times$ speedup compared to single-GPU execution, limited by the all-reduce communication pattern required for gradient aggregation.

