# OpenReview forum: "Curvature Meets Bispectrum: A Correspondence Theory for Transformer Gauge Invariants"
_ICLR.cc/2026/Conference — ICLR 2026 Conference Withdrawn Submission_

### Official Review · Reviewer_sFc4 · 2025-10-17

**Soundness:** 1
**Presentation:** 1
**Contribution:** 1
**Rating:** 0
**Confidence:** 4

**Summary:**

The paper studies identifiability in Transformers by describing the symmetries (“gauges”) of multi-head attention and claiming these account for the functional redundancies, with minor variants under architectures like GQA/MQA or RoPE. Building on this, it proposes two families of invariants to compare models up to those symmetries: a geometric construction based on Fisher–Rao curvature and an algebraic, permutation-robust bispectrum. The authors claim these two invariant views correspond in practice and report experiments across models with different numbers of heads to support the correspondence.

**Strengths:**

The identifiability of multi-head attention is an important and challenging problem, making the topic relevant and interesting.

**Weaknesses:**

Overall, I found the paper vague, confusing, and incorrect in several of its claims. In fact, I have serious concerns about the proofs of the main results. I will give two examples below, although this criticism applies to several claims and proofs in the paper.

- Theorem 2.1 claims to describe the full gauge group of a single layer of multi-head attention. This is a notoriously-challenging problem. The proof of this theorem is mis-referenced, and actually appears in Section A.4 (instead of A.14). Crucially, the proof of necessity/maximality – which is the non-trivial part of the claim – is just a few lines long, and is completely unclear to me. The crucial step is arguing that the only symmetries involving multiple heads are permutations. This is claimed to follow from assumption A1 (line 98), which is vague, informal, and confusing to me.

- Even further, Proposition 2.2 claims that the gauge group of a deep network factorizes as a product of layer-wise gauge groups. This is an even more challenging problem which, in my view, would probably require a highly-elaborate proof. Again, the provided proof is a few lines long (lines 619-624), and is completely vague, informal, and unclear.

The combination of mis-references, overall structure, and unsupported claims, raises the suspicion of heavy LLM involvement in the writing of the paper, including the mathematical statements and proofs. Moreover, a separate concurrent paper has been submitted to ICLR [1], where the claims, the notation, and the (incorrect) proofs are extremely similar to this work. The overlap between the two is substantial, supporting the suspicion that both works might contain significant portions of LLM-generated (technical) text.
There is a possibility I am misunderstanding the claims and the proofs, and that my suspicions are unfounded. However, at the current stage, I strongly believe that this work does not fit the standards of ICLR.





[1] Maximal Gauge Symmetry in Transformer Architectures. Under review at ICLR 2026. Available: https://openreview.net/pdf/47571adb71f14af6fe2b2392a8932faccaab6842.pdf

**Questions:**

In case my criticism is unfounded, I would be grateful to the authors if they could elaborate on the proofs of Theorem 2.1 and Proposition 2.2.

---

### Official Review · Reviewer_bmpP · 2025-10-29

**Soundness:** 2
**Presentation:** 2
**Contribution:** 2
**Rating:** 2
**Confidence:** 1

**Summary:**

The paper studies transformers from the following point of view:
If the Query and Key  matrices $W_Q$ and $W_K$ are multiplied by suitable matrices $A$ and $(A^{-1})^t$
so that $(W_QA)(W_K(A^{-1})^t)^t=W_QW_K^t$, the output of the transformer does not change.
This means that in the space of transformers there are operations which do not change the output.
Following the physical terminology, this means that there is a gauge group in the space of transformers.
As transformers are parametrized similarly to statistical models, one can consider the Fisher-Rao metric in
the space of transformers that makes the space a Riemannian manifold. This makes it possible to speak about curvature of the space. Moreover, as a gauge group operates in the space, one can consider transformers as a space with a bundle structure.  The paper seems to describe the gauge groups of the space of multihead transformers, it introduces curvature, studies holonomy (transport of objects along closed curves) and gives results for the bispectrum.

**Strengths:**

The paper connects deep and well developed mathematical results to machine learning. These findings may be valuable as it is interesting to understand transformers from a novel point of view.

**Weaknesses:**

The paper is difficult to read, at least for me. The paper would benefit a lot from an appendix that gives an introduction of to definitions and concepts.

**Questions:**

The paper appears to be interesting and personally I would very much like to learn more on the topic.
An appendix, starting from the definition of Fisher-Rao metric in the space of transformers, semi-direct products,
meaning of the bispectra (that has several different definitions in different areas of mathematics) would be very useful.
Also, an example of the concepts and results for a very simple, 1-layer transformer with 1 and 2 heads would most likely be very valuable
for many readers.

---

### Official Review · Reviewer_ADi5 · 2025-11-01

**Soundness:** 2
**Presentation:** 1
**Contribution:** 1
**Rating:** 2
**Confidence:** 1

**Summary:**

The paper establishes a connection between differential-geometric and harmonic-analytic invariants for characterizing functional equivalences in Transformer models. The authors derive the gauge structure for multi-head attention, showing that many parameter settings compute identical functions. They prove that Fisher-Rao curvature provides a lower bound for bispectral energy in a linearized neighborhood, with empirical validation. The work aims to bridge differential geometry and harmonic analysis to provide tools for identifying functional equivalence in transformers.

**Note:** This is out of my area of expertise, and this is indicated by my confidence score. My review is primarily concerned with the presentation of the paper and to the best of my understanding; I do not find myself qualified to assess the technical details.

**Strengths:**

- The paper addresses a mathematical characterization of parameter redundancies in Transformers, deriving the complete gauge structure for multi-head attention and establishing formal connections between two frameworks.

- The theoretical contribution appears solid with formal proofs provided.

- Empirical validation demonstrates high correspondence rates.

**Note:** This is out of my area of expertise, and this is indicated by my confidence score. My review is primarily concerned with the presentation of the paper and to the best of my understanding; I do not find myself qualified to assess the technical details.

**Weaknesses:**

-  The paper lacks clear motivation and practical relevance. While the authors establish a mathematical connection, they fail to articulate why this matters for the broader machine learning community or what concrete problems this solves. The only hint at application (line 415) mentions *"promising targets for head merging, shared-parameter updates, or careful learning-rate scheduling,"* but provides no elaboration, experiments, or demonstration of these applications.

- The presentation is inaccessible and poorly organized. The paper jumps between topics, primarily highly technical, without providing intuitive explanations or clear narrative flow. Even the introduction does not establish a clear high-level insight about the problem that the paper targets and its importance. For a venue like ICLR, the work should be comprehensible to the broader ML community.

- To my understanding, the experiments only validate the theoretical results in a simulation, rather than demonstrating their application (please correct me and clarify if this is wrong).

- The paper seems over-mathematicized without justification. Dense (and rather discrete) series of technical statements and theoretical machinery follow one after another, without first establishing why they are introduced and what they imply.


**Note:** This is out of my area of expertise, and this is indicated by my confidence score. My review is primarily concerned with the presentation of the paper and to the best of my understanding; I do not find myself qualified to assess the technical details. **That being said,** I stand by my assessment on the presentation. The paper's writing and organization need substantial improvement.

**Questions:**

1. Can the authors provide a concrete example where knowing the results derived on Fisher-Rao curvature and bispectral energy leads to a practical improvement or applicable understanding?

2. On line 415 the authors write *“This makes such layers promising targets for head merging, shared-parameter updates, or careful learning-rate scheduling.”* Can you elaborate and explain how?

---

### Official Review · Reviewer_vcLz · 2025-11-01

**Soundness:** 2
**Presentation:** 1
**Contribution:** 2
**Rating:** 2
**Confidence:** 3

**Summary:**

This paper shows a correspondence of geometric and algebraic invariants in analyzing transformer neural networks modulo their parameter symmetries. In Section 2, the authors introduce basic formalization of parameter symmetries of transformers. In Section 3, the authors introduce the curvature of Fisher-Rao mechanical connection as the geometric invariant under parameter symmetries and introduce a tractable numerical approximation scheme. In Section 4, the authors introduce the bispectrum arising from the head permutation symmetries after all continuous symmetries are removed through a choice of canonicalization as the algebraic invariant under parameter symmetries. The main results are in Section 5, showing that in stable regime of canonicalization, the squared curvature lower-bounds the bispectral energy. The authors provide an empirical validation of the result in Section 6, along with computational cost measurements showing that curvature is tractable while bispectrum is not.

**Strengths:**

S1. This paper tackles an important and potentially impactful problem of studying invariants of the transformer architecture modulo parameter symmetries. I am unaware of prior work that takes the approach of linking geometric and algebraic invariants for the problem.

**Weaknesses:**

W1. I am unsure how the main result in Theorem 5.1 is practically relevant, that is, how specifically it can be used to understand or analyze transformers in practice. It does show that one invariant quantity lower-bounds another, and experiments do seem to validate this inequality, but in which specific ways can this be useful? Figure 1 suggests to some degree that the slope constant roughly stays in some numerical range, which might be useful if estimating the bispectrum is desired in some context, but this is not guaranteed by the main result (Theorem 5.1) and could break under different empirical setups such as different choices of canonicalization.

W2. The readability of the paper could benefit from revising the writing for readers unfamiliar with the used tools.

W3. I was unable to capture how the main symmetry results in Section 2 significantly add upon already known results in the literature [1, 2].

[1] Silva et al., Hide & Seek: Transformer Symmetries Obscure Sharpness & Riemannian Geometry Finds It, ICML 2025.

[2] Ziyin, Symmetry Induces Structure and Constraint of Learning, ICML 2024.

**Questions:**

I have no particular questions but would like to hear the authors' response on the weaknesses.

---

### Note · Authors · 2025-11-12

**Comment:**

I learned that a revision of the paper was accepted in a neurips workshop on 12/7/25. so I will present the work at that workshop.

Thanks to the reviewers for the great feedbacks.

**Withdrawal Confirmation:**

I have read and agree with the venue's withdrawal policy on behalf of myself and my co-authors.